# Analysis of Experimental Data on Changes in Various Structures and Functions of the Rat Brain following Intranasal Administration of Fe_2_O_3_ Nanoparticles

**DOI:** 10.3390/ijms24043572

**Published:** 2023-02-10

**Authors:** Ilzira A. Minigalieva, Yuliya V. Ryabova, Ivan G. Shelomencev, Lev A. Amromin, Regina F. Minigalieva, Yuliya M. Sutunkova, Larisa I. Privalova, Marina P. Sutunkova

**Affiliations:** 1Yekaterinburg Medical Research Center for Prophylaxis and Health Protection in Industrial Workers, 30 Popov Street, 620014 Yekaterinburg, Russia; 2Laboratory of Stochastic Transport of Nanoparticles in Living Systems, Laboratory of Multi-Scale Mathematical Modeling, Ural Federal University, 51 Lenin Avenue, 620002 Yekaterinburg, Russia

**Keywords:** neurotoxicity, kinetics of nanoparticles, electron microscopy

## Abstract

Particulate matter, including iron nanoparticles, is one of the constituents of ambient air pollution. We assessed the effect of iron oxide (Fe_2_O_3_) nanoparticles on the structure and function of the brain of rats. Electron microscopy showed Fe_2_O_3_ nanoparticles in the tissues of olfactory bulbs but not in the basal ganglia of the brain after their subchronic intranasal administration. We observed an increase in the number of axons with damaged myelin sheaths and in the proportion of pathologically altered mitochondria in the brains of the exposed animals against the background of almost stable blood parameters. We conclude that the central nervous system can be a target for toxicity of low-dose exposure to Fe_2_O_3_ nanoparticles.

## 1. Introduction

The health effects of inhalation exposure to ultrafine particles in industrial workers have not been studied sufficiently so far. Numerous metallurgical processes and electric arc welding generate aerosols of iron condensation, containing submicron and nano-sized particles, the proportion of which can be as high as 40% of PM_2.5_ [1]. We assume that inhalation of such particles could be more detrimental to human health than expected based on mass measurements [2] in the absence of dispersion modeling.

Up-to-date literary sources describe multiple organ toxicity of iron oxide nanoparticles. The impact of iron-rich airborne particles on the cardiovascular system has been proven both in vivo using Danio rerio [3] and in epidemiological studies [4]. Adverse effects of iron oxide nanoparticles on the reproductive system, thyroid gland, and organs of the mononuclear phagocytic system are also known to date [5]. Toxicity of iron-rich nanoparticles has been shown in vitro for endothelial cells of the human umbilical vein of the HUVEC line [6], hepatomas of SK-Hep-1 and Hep3B lines [7], and neuroblastomas of the SHSY-5Y line [8], although the cytotoxic effect of these particles could not always be established unambiguously [9]. The ability of 2.3 and 4.2 nm Fe_3_O_4_ nanoparticles to induce oxidative stress, owing not only to the small size but also to their chemical identity, has been also demonstrated in vivo [10]. At the same time, physical parameters of iron oxide nanoparticles have a significant effect on cellular uptake, cytotoxicity, and their distribution and clearance [11].

Recent studies have proven that the central nervous system is one of the targets for toxicity of metal and metal oxide nanoparticles [12,13,14], which can penetrate into the brain by crossing the blood–brain barrier or by retrograde transport through the olfactory nerve endings [14,15,16,17].

The neurotoxic effect of iron nanoparticles, including those of Fe_2_O_3_, has been repeatedly evidenced by animal experiments. In their studies on mice exposed to Fe_2_O_3_ nanoparticles at doses of 25 and 50 mg/kg body weight for four weeks, Dhakshinamoorthy et al. showed a change in motor behavior and spatial memory, as well as impaired permeability of the blood–brain barrier and accumulation of nanoparticles in the brain. The authors associated the observed health effects with the accumulated nanoparticles, oxidative stress, and apoptosis in the brain of mice [18].

Maher B. et al. also associated the exposure to iron compounds (magnetite nanoparticles) with a unique combination of enhanced production of reactive oxygen species, surface charge, and strong magnetism [19].

Wu J. et al. [20] conducted a study with acute and subacute intranasal exposure of Sprague Dawley male rats to Fe_2_O_3_ nanoparticles and found the ability of the latter to deposit in brain tissues and damage the striatum [20]. Oral exposure to iron nanoparticles at a total dose of 5 mg/kg induced an increasing swelling of brain tissues observed histologically [21].

The kinetics of Fe_2_O_3_ was evaluated by Garcia-Fernandez J. et al. in in vitro studies by modeling the distribution of ultrasmall Fe_2_O_3_ particles in the gastrointestinal tract [22]. Popescu R. et al. showed that Fe_3_O_4_ could be captured by cells of tissue macrophages in the liver, spleen, and lungs [23,24]. Long-term studies conducted by Soares G et al. demonstrated poor clearance of nanoparticles and their increased accumulation in the liver and spleen [25]. In addition, nanoparticles of the same elemental composition but different sizes have unequal accumulation parameters [26]. At the same time, we have failed to find open access articles demonstrating the kinetics of iron oxide nanoparticles in brain structures.

The studies that we have found were conducted using relatively high doses of Fe_2_O_3_ nanoparticles. Current occupational exposure doses, however, demonstrate an obvious decreasing trend, e.g., in Canada [27], the United Kingdom [2], and the Russian Federation [28]. Data are accumulating on the dangers of low concentrations of nanoparticles (<1 μg/mL) in studies on Danio rerio [29]. Our objective was to establish whether Fe_2_O_3_ nanoparticles are capable of exerting toxic effects at low doses. Based on published data, we assumed that even low-dose exposure to iron oxide nanoparticles can induce changes at the ultrastructural and, probably, organismal level (behavioral reactions).

## 2. Results

### 2.1. Analysis of Experimental Data on General Toxic Effect

The analysis of the rats’ behavioral activity data revealed some statistically significant changes in different weeks of the exposure period (Figure 1). Total physical activity changed in the first place (Figure 1a): a pronounced downward trend was noted in each study group in weeks one to three, followed by a rise. However, while in the control group, locomotion recovered to the initial level and even exceeded it, that of the exposed animals demonstrated a pronounced decrease by the end of the experiment. In addition, while exploratory behavior, as judged by the number of head-dipping occurrences, in the control animals remained relatively stable, it dropped sharply in the exposed rats (Figure 1b).

Anxiety in the exposed animals was measured in the hole-board test by grooming (Figure 2a) and defecations (Figure 2b). According to the data obtained, the rats experienced some stress only in the third week of exposure, which leveled off by the end of the exposure period.

The reaction time of the rats to a stimulus differed significantly across the exposure period, as evidenced by changes in the summation threshold index shown in Figure 3. It is worth noting that statistical changes were observed already in week 1, which subsequently leveled off by the end of the experiment.

Due to the ambiguity of data on changes in behavioral reactions, we decided to analyze average indicators of behavior and summation threshold index, as shown in Table 1.

Despite the pronounced and unambiguous changes in the behavior of the exposed animals, we observed no significant changes in weight or hematological parameters. We can only note a statistically insignificant yet obvious increase in serum myelin basic protein (MBP) (2.99 ± 0.60 in the exposed animals against 2.44 ± 0.43 in the controls).

The analysis of the proportions of different cells in rhino-cytological wash specimens (Figure 4a) and imprint smears (Figure 4b) of the nasal cavity in both groups showed their comparability. Yet, the number of lymphocytes in eosinophils and macrophages demonstrated a slight rising trend.

### 2.2. Electron Microscopy Findings

Electron microscopy revealed deposits of Fe_2_O_3_ nanoparticles in tissues of olfactory bulbs of the rats’ brains; their composition was later confirmed by energy-dispersive X-ray spectroscopy (Figure 5a,b). Electron-dense objects containing Fe were not found in the samples from the control animals; no background signal was detected either.

Such an approach enables abstraction from the authors’ judgment about the objects observed and replaces them with objective EDS images when analyzing in high resolution. The method applied can only confirm the presence of nanoparticles in tissues but cannot prove their absence due to its low sensitivity.

We used electron microscopy to study the axons of the neurons in two parts of the rats’ brains for ultrastructural damage of myelin sheaths. The most common abnormality of the myelin sheath can be described as multiple-crater formations of different diameters and circumferences (Figure 6).

The examination of the ultrastructure of the myelin sheath showed an increase in the number of axons with the damaged myelin sheath in tissues of basal ganglia of 32.6% following the exposure to Fe_2_O_3_ NPs compared to the controls (Table 2).

We noticed a trend toward an increase in the proportion of axons with damaged myelin sheaths in the olfactory bulb; however, this pattern could not be statistically confirmed. There were also no significant differences in this indicator between brain regions within one group.

The mitochondrial profile was examined by electron microscopy according to the classification proposed by Sun et al. [30], based on the topology of the inner mitochondrial membrane: normal, normal vesicular, vesicular, swollen vesicular, and swollen. The analysis of electron microscopy images confirmed the presence of all five morphotypes of mitochondria in brain neurons in both groups of rodents (Figure 7).

Mitochondria were classified by two independent readers, and then, the distribution of mitochondrial morphotypes was assessed for each group of animals by brain regions (Figure 8).

Using Pearson’s goodness-of-fit test (χ^2^), we confirmed the abovementioned differences in the distribution of mitochondrial morphotypes between the exposed animals and the controls (χ^2^BG (4; 0.05) = 1090.6; *p* ≤ 0.0001; χ^2^OB (4; 0.05) = 1010.0; *p* ≤ 0.0001).

Detailing the alterations in the distribution of mitochondria by morphotypes by pairwise comparison of the Mann–Whitney U test reaffirmed that the proportion of mitochondria with the normal morphotype in tissues of basal ganglia of the exposed animals was 26.4% lower than that in the controls. The proportions of mitochondria of the normal vesicular and vesicular morphotypes were 31.9 and 10.0% higher in the exposed rats compared to the controls, respectively. In tissues of olfactory bulbs of the exposed animals, the proportions of mitochondria of the normal and swollen vesicular morphotypes were 36.4 and 4.9% lower than in the controls, respectively. The shares of mitochondria of the normal vesicular and vesicular morphotypes were 19.8 and 21.8% higher in the exposed rodents, respectively. In general, the distribution of mitochondria by morphotypes in the exposed animals was shifted toward the vesicular morphotype, with a predominance of the normal vesicular type compared to the controls.

We also established distinctions in the distribution of mitochondrial morphotypes between different brain regions in both groups (χ^2^control (4; 0.05) = 283.1; *p* ≤ 0.0001; χ^2^Fe_2_O_3_ NPs (4; 0.05) = 183.3; *p* ≤ 0.0001).

In the tissues of olfactory bulbs of the control animals, the proportion of mitochondria of the normal morphotype was 20.6% lower relative to the basal nuclei, and that of the normal vesicular one was 17.6% higher. In the rodents exposed to Fe_2_O_3_ NPs, we observed a similar trend toward a decrease in the proportion of normal mitochondria and an increase in that of mitochondria with damaged topology of the inner membrane in tissues of olfactory bulbs compared to basal nuclei. The percentage of mitochondria of the normal morphotype in tissues of olfactory bulbs was 14.7% lower; in contrast, the percentage of the vesicular morphotype was 12.1% higher.

However, no significant differences were found between mitochondria of the vesicular morphotype due to a high variability of values (*p* = 0.160; Mann–Whitney U test). The results of the paired Student’s *t*-test were similar for all the comparisons.

## 3. Discussion

We established the presence of nanoparticles in tissues of olfactory bulbs but not in basal ganglia. This finding is consistent with the theory of migration of nanoparticles to the brain along the olfactory nerve following inhalation exposure. This feature was noted for a number of metal nanoparticles [31,32]. Such a pattern of penetration of nanoparticles to brain tissues may be due to the advantage of the olfactory pathway where the blood–brain barrier is the most permeable [14,15]. Oberdörster et al. (in their study using, ca., 36-nm particles) concluded that the most likely mechanism of their penetration was deposition on the olfactory mucosa of the nasopharyngeal region of the respiratory tract and subsequent translocation through the olfactory nerve [33]. Maher et al. proved their conclusions in a large epidemiological study by establishing that magnetite nanoparticles could enter the brain directly through the olfactory nerve [19].

The absence of nanoparticles in basal ganglia can be a false negative due to the low sensitivity of the method in combination with a small number of Fe_2_O_3_ nanoparticles that have penetrated into basal ganglia. Fe_2_O_3_ might not be detected in basal ganglia due to changes in their physicochemical properties in the body, i.e., a possible formation of a protein corona [34] not assessed in our study. For instance, de Oliveira G et al. demonstrated a significant difference between the effects of dextran-coated IONPs (CLIO-NH2) and uncoated IONPs (UCIO) in zebrafish larvae [29].

It is worth noting that Fe_2_O_3_ NPs were found both in subcellular structures (e.g., cell nuclei and mitochondria) and in neuron cytoplasm. Our results are consistent with the findings of previous studies showing the presence of nanoparticles in intracellular structures, including mitochondria and nuclei, despite the differences in physical (shape and size) and chemical (silver and gold) characteristics of nanoparticles [35,36]. It is impossible to unambiguously conclude the mechanisms of penetration of Fe_2_O_3_ nanoparticles to subcellular structures, since the true mechanism of the toxic effect of nanoparticles may be the result of several risk factors acting simultaneously and difficult to differentiate (particle charge, shape, size, coating, chemical identity, and cellular uptake rate) [37].

Intranasal administration of Fe_2_O_3_ nanoparticles caused significant changes in behavior but not in cytological characteristics of nasal wash specimens and imprint smears of the nasal cavity, hematological parameters, or biochemical parameters of blood serum, with the exception of a statistically insignificant increase in the serum level of the myelin basic protein of 22.5%. Our findings are consistent with those reported by Badman et al. [38], proving that iron nanoparticles can be harmful to neurons at significantly lower concentrations than previously reported for other types of cells. We assume that we observed early signs of toxic effects at the organismal level [37]. The higher level of the myelin basic protein, the second most abundant protein in the myelin of the central nervous system [38] and a known marker of brain tissue damage [39], can give evidence of the damaged myelin sheath of axons. This damage was, in its turn, confirmed by electron microscopy results showing 1.6- and 1.4-fold increases in the number of axons with the damaged myelin sheath in tissues of basal ganglia (*p* < 0.05) and olfactory bulbs, respectively (Figure 5, Table 2). A decrease in the behavioral activity of the exposed animals, especially reduced locomotion, head dipping, and rearing (Figure 1 and Figure 2, Table 1), may indicate nerve conduction disorders in the central nervous system. Yet, the ability of the nervous system to sum up subthreshold impulses judged by the summation threshold index remained the same (Figure 2). Thus, damage to the myelin sheath related to exposure to Fe_2_O_3_ NPs in this case did not affect the nerve conduction velocity; we assume that either the damage was compensated for or our methods failed to detect “subtle” changes. Other components could potentially contribute to behavioral changes in the rats. Given the importance of mitochondrial function for neuronal health, interactions with nanoparticles can be detrimental to the nervous system. Mitochondrial damage could be immediately caused by the effect of Fe_2_O_3_ NPs on brain structures, which we assume following the detection of nanoparticles in them (Figure 4). The potency of such an effect has been previously demonstrated for silica [40], silver [41], and gold [42] nanoparticles. Induction of oxidative stress specific for most nanoparticles [43,44,45] could not be excluded either, since mitochondria are important targets for almost all types of damaging agents, including stress [46]. The damage to mitochondria could occur indirectly under the effect of dissolved iron in ionic form [46]. We cannot rule out the contribution of both nanoparticles and ferric ions to this damage.

## 4. Materials and Methods

### 4.1. Experimental Data

Within the framework of the present work, we have analyzed data obtained in a subchronic toxicity study of Fe_2_O_3_ nanoparticles (Fe_2_O_3_ NPs) sized 18 ± 4 nm in the total dose of 0.45 mg Fe_2_O_3_ NPs per rat, administered intranasally to three- to four-month-old female rats weighing about 200 g, with 12 animals per group. The choice of dose in the study we analyzed was limited by the nanoparticle concentration in the suspension and the maximum allowable volume for intranasal instillation in rats. The design of the experiment was similar to that described elsewhere [47]. The sample size was decided based on the experience acquired in our previous studies and the assumed minimum number of rodents necessary to obtain trustworthy results. Suspensions of Fe_2_O_3_ nanoparticles were prepared at the Ural Federal University, Yekaterinburg, Russian Federation, by pulsed-laser ablation of thin sheet targets of 99.9% pure iron in sterile deionized water, as described elsewhere [48]. We examined the results of the hole-board test used for assessing exploratory behaviors in rodents (head dips into holes and sniffing, locomotion, grooming, and defecations) and summation threshold index. To adjust for the bias caused by adaptation to stress related to injections, we averaged the hole-board test results and the summation threshold index for six weeks. We also examined the results of the rhino-cytological study of nasal wash specimens [49] and imprint smears of the nasal cavity [50] for seven animals. In addition, we considered the results of autopsies, general blood tests, biochemical tests (including the level of myelin basic protein), and electron microscopy images showing the severity of damage to both the myelin sheath of axons and mitochondria. Randomization of the visual fields and sample preparation for electron microscopy are shown in Figure 9.

Energy dispersive X-ray spectroscopy (EDS) spectra were analyzed using the AZtec software (Oxford Instruments, Abingdon, UK).

To identify Fe_2_O_3_ nanoparticles in rat brain structures, 420 ultrathin sections of two parts of the brain of the exposed and control animals were examined.

When analyzing morphological changes in the myelin sheaths of axons and assessing the state of neuron mitochondria, a series of five to six sections was prepared, and the highest quality ultrathin section of the series was divided into 20+ fields of view, evenly distributed over the entire surface, and then examined.

When determining the degree of damage to the myelin sheath of axons in sections of two regions of the brain, 1387 unique sites of the myelin sheath of axons (40 to 60 sites × 2 areas of the brain × 7 rats × 2 groups) were ranked by the presence or absence of damage to its structure.

When analyzing the damage to mitochondria following the exposure to nanoparticles, we assumed five successive stages of transformation of the inner membrane according to the classification of Sun et al. [30]. We ranked 4490 mitochondria in 84 samples from 14 rats of the control (*n* = 7) and experimental (*n* = 7) groups. Morphotyping was carried out by two independent readers to exclude bias in establishing mitochondrial morphotypes, after which the results of the percentage distribution of morphotypes were averaged for each type of tissue and animal.

### 4.2. Statistical Data Analysis

The statistical significance of data on behavioral reactions and blood parameters was established using the Student’s *t*-test. In addition, we compared *p* values estimated using the Student’s *t*-test and the Mann–Whitney U test, and their general coincidence proved the appropriateness of applying the *t*-test.

Scanning electron microscopy findings were analyzed using the Statistica 12 software. The statistical significance of differences in the distribution of mitochondrial profiles was assessed using Pearson’s goodness-of-fit test (χ^2^), while that between the groups was determined using the Mann–Whitney U test and Student’s *t*-test.

The difference between means was considered statistically significant if the probability of its occurrence by chance was equal to or less than 0.05 (*p* ≤ 0.05).

## 5. Conclusions

Our data analysis showed the ability of Fe_2_O_3_ nanoparticles at a total dose of 0.45 mg to have a toxic effect on rats. Administered intranasally, the particles significantly affected behavioral reactions but induced no changes in cytological characteristics of nasal swabs or imprint smears of the nasal cavity, hematological parameters, or biochemical indices of blood serum.

The examined kinetics demonstrated the ability of nanoparticles to enter the brain structures and to distribute unevenly in different parts of the rat’s brain. This conclusion requires further research to elucidate patterns of accumulation of Fe_2_O_3_ nanoparticles in the brain and to establish the dependence of their distribution therein on physical properties, dose, exposure duration, and route of administration of the nanoparticles. In addition, it is necessary to study correlations between the presence of nanoparticles in tissues, cells, and organelles, and ultrastructural damage to biological objects under study.

## Figures and Tables

**Figure 1 ijms-24-03572-f001:**
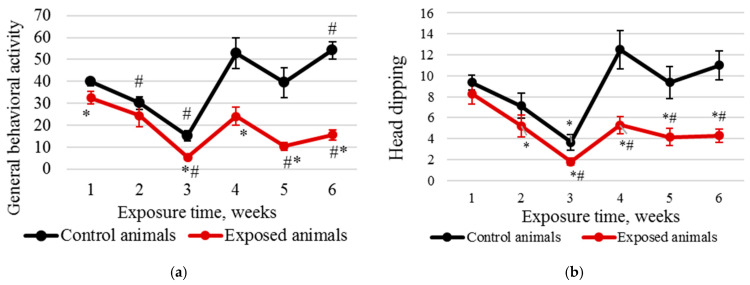
Results of the hole-board test: (**a**) general behavioral activity decreased by the end of week 6 in the exposed animals and increased in the controls; (**b**) number of head dips into holes sharply reduced by the end of the experiment in the exposed animals and was stable in the controls. The abscissa shows the exposure time in weeks, and the ordinate shows the value of the corresponding indicator; * compared with the controls; # compared with the value in the same group in week 1.

**Figure 2 ijms-24-03572-f002:**
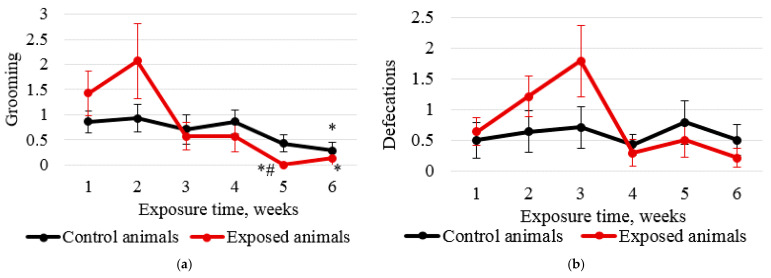
Hole-board test results: (**a**) grooming; (**b**) defecations. Based on the graphs, the rats experienced some stress in week 3 of intranasal injections of Fe_2_O_3_ or deionized water, while the level of anxiety of exposed animals in week 4 and until the end of the experiment was lower than that of animals in the control group. The abscissa shows the exposure time in weeks and the ordinate shows the value of the corresponding indicator; * compared with the controls; # compared with the value in the same group in week 1.

**Figure 3 ijms-24-03572-f003:**
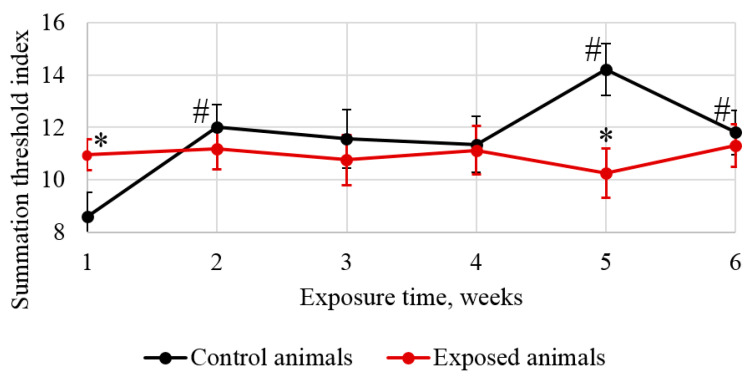
Changes in the summation threshold index. The reaction time of the rats to a stimulus differed significantly across the exposure period. The abscissa shows the exposure time in weeks, and the ordinate shows the value of the corresponding indicator; * compared with the controls; # compared with the value in the same group in week 1.

**Figure 4 ijms-24-03572-f004:**
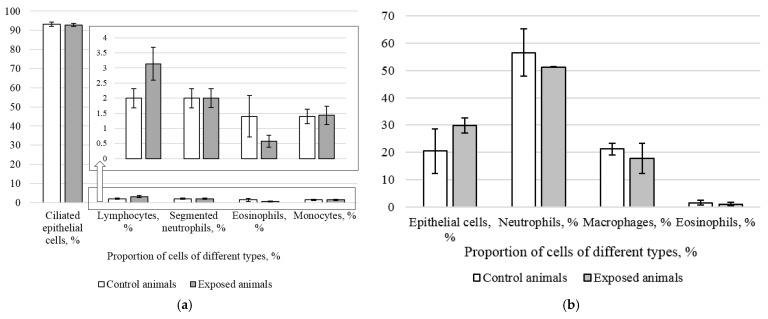
Proportion of cells of different types in rhino-cytological wash specimens (**a**) and imprint smears (**b**) of the nasal cavity in the exposed and control rodents.

**Figure 5 ijms-24-03572-f005:**
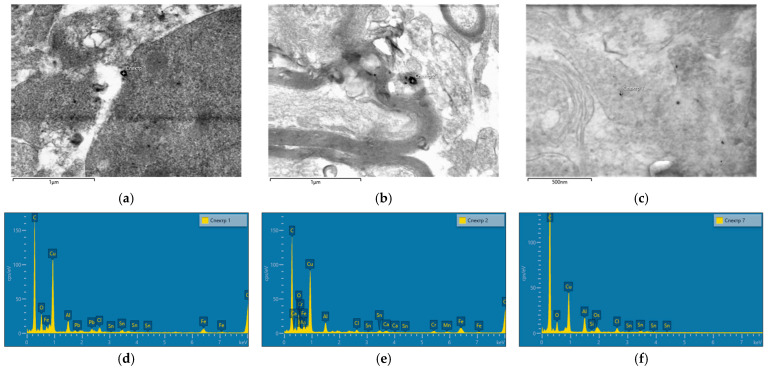
(**a**,**b**) STEM images of the olfactory bulb tissues with deposits of Fe_2_O_3_ nanoparticles and (**c**) without them in the control rodents; (**d**–**f**) EDS spectra of the regions marked on image (**a**), (**b**) and (**c**), respectively.

**Figure 6 ijms-24-03572-f006:**
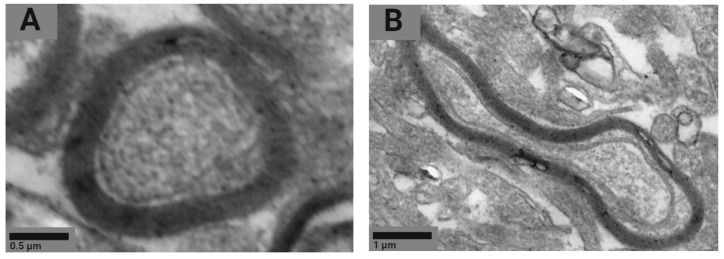
Representative STEM images of the normal (**A**) and damaged (**B**) myelin sheath.

**Figure 7 ijms-24-03572-f007:**
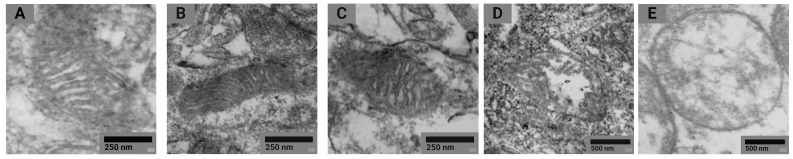
Representative STEM images of normal (**A**), normal vesicular (**B**), vesicular (**C**), swollen vesicular (**D**), and swollen (**E**) mitochondrial morphotypes found in animal tissues in both groups of rats.

**Figure 8 ijms-24-03572-f008:**
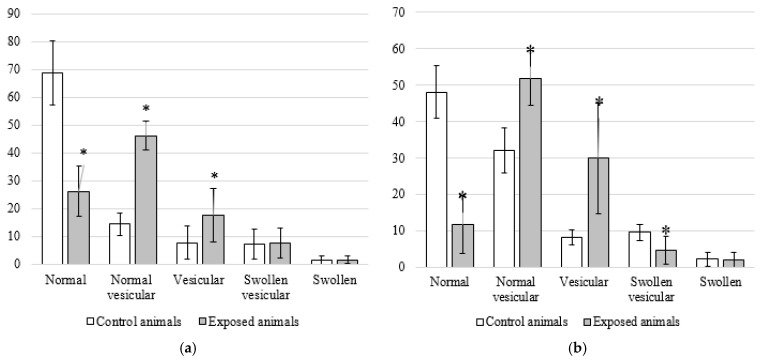
Damage to the ultrastructure of neuronal mitochondria in the animals exposed to Fe_2_O_3_ nanoparticles and controls by brain regions: (**a**) basal ganglia and (**b**) olfactory bulb. Notes: values are given as the arithmetic mean of the percentage of mitochondria in the total number of detected neuronal mitochondria; * *p* < 0.05 (Mann–Whitney U test) compared with the controls.

**Figure 9 ijms-24-03572-f009:**
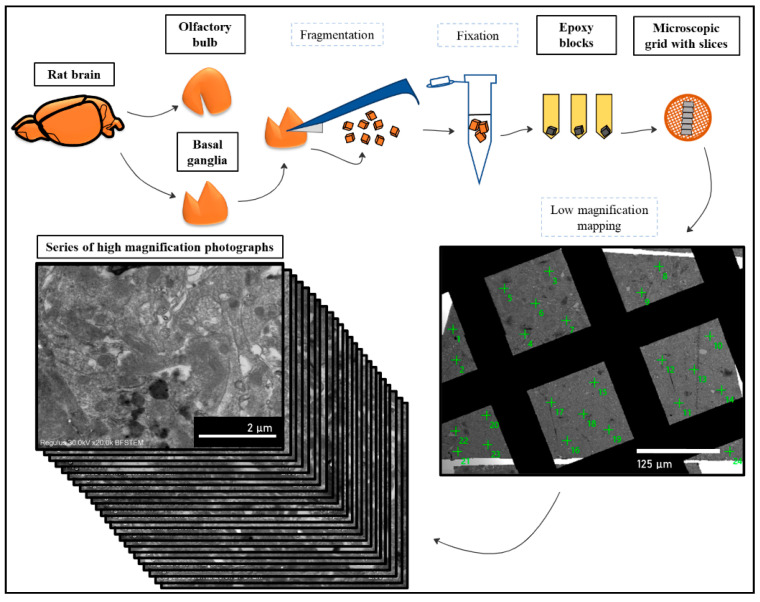
Sampling and sample preparation for electron microscopy.

**Table 1 ijms-24-03572-t001:** Average indicators of behavioral reactions of rats in the hole-board test.

Variables	Control Group (Deionized Water)	Exposure Group (Fe_2_O_3_ NPs)
Summation threshold index	11.63 ± 0.43	10.93 ± 0.34
Total activity (except defecations)	38.58 ± 2.34	18.70 ± 1.62 *
Head lifting	6.94 ± 0.93	3.60 ± 0.66 *
Rearing	0.86 ± 0.15	0.21 ± 0.06 *
Locomotion	12.54 ± 0.88	6.00 ± 0.52 *
Head dipping	8.83 ± 0.60	4.83 ± 0.38 *
Hole sniffing	8.74 ± 0.77	3.26 ± 0.52 *
Grooming (5 s—1 score)	0.68 ± 0.09	0.80 ± 0.18
Defecations	0.60 ± 0.12	0.77 ± 0.14

* *p* ≤ 0.05 (Student’s *t*-test), compared with the control group.

**Table 2 ijms-24-03572-t002:** Damage to the ultrastructure of myelin sheaths of axons in two parts of the brain in the rats exposed to Fe_2_O_3_ nanoparticles and the control animals.

Part of Brain	Damaged Areas of the Myelin Sheath (Mean ± 95% CI), %	
	Control Group	Exposure Group
Olfactory bulb	57.0 ± 9.4	81.6 ± 21.5
Basal ganglia	50.9 ± 8.1	83.5 ± 9.7 *

Values are given as the proportion of axons with damaged sheaths of the total number of axons detected; * *p* < 0.05 (Mann–Whitney U test) compared with the controls.

## Data Availability

The data presented in this study are available on request from the corresponding author.

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
