# Peer review of "Analysis of Experimental Data on Changes in Various Structures and Functions of the Rat Brain following Intranasal Administration of Fe2O3 Nanoparticles"

_ijms, 2023, doi:10.3390/ijms24043572_

Round 1

Reviewer 1 Report

This is a report on an experimental study to explore the changes in various structures and functions of the rat brain following intranasal instillation of Fe2O3 nanoparticles. The evidence includes some behavior assessments including general behavior and anxiety responses, and morphological changes in the myelin sheaths of axons and assessing the state of neuron mitochondria. The topic is very emerging given the health concern of the air pollutants, within which iron particles are one of the major contributors. There were some very interesting results, however, the report will benefit from revisions taking the following remarks into account.

1.     This study only used one dose of Fe2O3 for intranasal instillation. Please add more information in terms of how this dose is selected. Also, please add some more information in the Discussion how the results in this study in comparison to other studies in terms of the biological endpoints.

2.     Figure 4 legends don’t seem fit with the figure. Please correct.

3.     This study include rhino-cytological wash and imprint smears followed with cell population analysis. The size of the Fe2O3 used in this study is 18±4 nm. It is generally accepted that nano-sized particles deposited most in the deep lung. Please explain why bronchial lavage with cell pollution analysis was not selected.

4.     For Figure 6 and Table 1, it is said the examination of the ultrastructure of the myelin sheath showed an increase in 128 the number of axons with the damaged myelin sheath in tissues of basal ganglia by 32.6 129 % following the exposure to Fe2O3 NPs compared to the controls. Please explain more in details on how the ultrastructure of myelin sheaths of axons is examined. For example, was the person blind to treatment and different groups? how many areas per rat were examined, how

5.     Figure 5 is currently labelled as Figure 1. Please correct.

6. In the conclusion, it is said that “this study demonstrated that nanoparticles to get into the brain structures and to distribute unevenly in different parts of the rat’s brain”. As 420 sections of the rat brains have been examined within this study, could any pattern of accumulation of Fe2O3 nanoparticles in the brain be elucidated from this study? Have the authors thought of further analyzing this?

Author Response

Dear Reviewer,

We appreciate your comments and suggestions for improvement. Please find the answers to your questions and/or remarks below.

  1. We used only one dose because we were limited by the maximum concentration of nanoparticles and the volume of the suspension instilled intranasally. We might have used a lower dose but decided otherwise. We were afraid of observing no toxic effect while making the rats suffer. Since we had no task to establish/specify LOAEL or NOAEL for Fe2O3 nanoparticles, only one dose was chosen for the study. We have added the following explanation to the article: “The choice of the dose in the study we analyzed was limited by the nanoparticle concentration in the suspension and the maximum allowable volume for intranasal instillation in rats.”

We have also supplemented the Introduction and Discussion sections with appropriate citations and highlighted all the additions in yellow.

  1. Figure 4 captions have been corrected.
  2. Of course, we agree with you that nano-sized particles deposit most in the deep lung. Yet, the purpose of our study was to establish their entry/penetration into the brain. This is the reason why we present no data on bronchoalveolar lavage (this was the subject of another study published elsewhere in which we established alveolar phagocytosis in response to instillation of the same suspension).
  3. We have specified how the ultrastructure of the myelin sheath of axons was examined: “When determining the degree of damage to the myelin sheath of axons in sections of two regions of the brain, 1,387 unique sites of the myelin sheath of axons 60 sites × 2 areas of the brain × 7 rats × 2 groups) were ranked by the presence or absence of damage to its structure.”
  4. The number of Figure 5 has been corrected.
  5. Within the framework of the present study, we cannot determine clear patterns in the distribution of Fe2O3 nanoparticles. We assume that (1) additional experiments with labeled nanoparticles (radioactive labels, for instance, can help differentiate biological iron from that administered) or (2) supplementary mathematical processing, construction of models of stochastic transport of nanoparticles in the brain, are needed to reveal them.

Reviewer 2 Report

The article is incomplete and poorly written. The quality of the research is very low. There is no justification for the sample size of the animals. The results are presented unacceptably. Figures 1-3 are mislabeled. It is not known what the trend is. There is no statistical analysis of the results obtained. The authors did not present testing the significance of the obtained results or building confidence intervals of the obtained values. The error bars are the same in every chart, why? The STEM images do not have the most important structures marked. The EDS/EDX test is burdened with a very large error, so the interpretation of these results is highly questionable. The discussion of the results is incomplete. There is no reference to similar works appearing in the last two years. There are no clearly formulated research hypotheses, which is why it is difficult to refer to the conclusions. The article is not thematically relevant to the magazine.

Author Response

Dear Reviewer,

We appreciate your valuable comments on the manuscript. Please find the answers to your questions and/or remarks below.

1) The choice of the animal sample size was based on provisions of OECD TG 413 and the design of our experiment. It was difficult to predict variability of the parameters under study and we were therefore unable to estimate the appropriate sample size using the Power and Sample Size Analysis. It should also be taken into account that no method for determining the sample size is perfect [doi.org/10.1177/0023677217738268]; thus, we decided on the sample size based on the experience acquired in our previous studies. The manuscript has been supplemented with the following: “The sample size was decided on based on the experience acquired in our previous studies and the assumed minimum number of rodents necessary to obtain trustworthy results.”

2) The mistake has been corrected.

3) Statistical analysis is described in par. 4.2. Would you please clarify what should be specified?

4) The mistake has been corrected and the figures amended.

5) We deliberately avoided marking the most important structures in the STEM images in order to focus attention of the readers on the objects under study and not to overload the images with descriptions unrelated to the issue.

6) Of course, the EDS/EDX technique is noted for certain ambiguity and this fact is mentioned in our manuscript as follows: “The absence of nanoparticles in basal ganglia can be false negative due to the low throughput of the method in combination with a small number of Fe2O3 nanoparticles that have penetrated to the basal ganglia.” Yet, the EDX analysis in the control group showed no traces of Fe (neither electron dense inclusions, nor the background signal of the instrument or the matrix of samples/sections) making us believe that detection of Fe2O3 NPs in olfactory bulbs of the exposed animals was reliable. We have added the EDS spectrum for the control group to prove the absence of a signal from essential (biological) iron and from STEM tool contamination. The Results section has been supplemented with the following: “Electron dense objects containing Fe were not found in the samples from the control animals; no background signal was detected either.”

7) The manuscript has been revised and supplemented with citations from recent similar publications, all highlighted in yellow. Please advise if we should pay attention/focus on works of specific authors.

8) The research objective was formulated in the introduction as follows: “Our objective was to establish whether Fe2O3 nanoparticles are capable of exerting toxic effects at low doses.” To clarify the hypothesis, we have added the following: “Based on published data, we assumed that even a low dose exposure to iron oxide nanoparticles can induce changes at the ultrastructural and, probably, organismal level (behavioral reactions).”

Reviewer 3 Report

The paper supports recent major work about potential toxicity of iron oxide nanoparticles and has relevance for both medical and fundamental scientific work, given that they have found effects at lower doses especially. Overall the authors seem to have done a careful and fair assessment of their results. I do recommend some improvements to the statistical presentation especially:

Figures 1-3 and Table 1 the error bar types are not specified. Please use 95% CI around the mean and report it in the text. Standard error (SEM) is too hard for readers to interpret if that is what you used. 

Given the p-values in Table 1 the results seem significant enough though, just improve the presentation.

When you did the t-tests, did you use repeated measures ANOVA for the fact that this is the same rats (I think) being measured over time?

For "We can only note a statistically insignificant, yet obvious increase in serum myelin basic protein (MBP) (2.99 ± 0.60 in the exposed animals against 2.44 ± 0.43 in the controls)." again please use 95% mean CI. Does MBP fluctuate a lot within a single individual rat over time or does it have much different baselines per rat but small temporal variation within rats? Please clarify in the text. The effect may be different if you control for different baselines. For the exposed mice did you see the before and after exposure MBP values at the single rat level? Please comment if so and if not, comment whether this is a useful follow-up direction for future work. If so the distribution of (after - before) per mouse levels may be worth analyzing.

For " total number of axons " in the second Table 1 (you have two Table 1's, please correct), can you report the number of axons measured in total, and the average number of axons measured per rats +/- 95% CI? I see you've done the former for mitochondria but mean per animal +/- 95% CI also would help in each case including for mitochondria. 

Your second Figure 1 is Figure 5 I think. For the STEM pictures, please include a control slide showing the Fe signals in control mice. There is a small possibility the signal could be due to biological Fe naturally in the body or from STEM tool contamination. It would be helpful to know how much of the Fe signal is from potential noise sources.

All iron oxide particles are not equal. There can be different forms of the iron oxide itself (e.g. magnetite), and small changes in size, clumping, or surface properties can significantly affect the uptake and toxicity profile of iron oxide nanoparticles. Can you include some brief discussion of this to better inform and warn readers? These papers have good discussion in that area and could be cited and briefly discussed, as they are major recent papers in your area of research:

de Oliveira GMT, de Oliveira EMN, Pereira TCB, Papaleo RM, Bogo MR. Implications of exposure to dextran-coated and uncoated iron oxide nanoparticles to developmental toxicity in zebrafish. J. Nanoparticle Res. 2017

Oberdorster G, et al. Translocation of inhaled ultrafine particles to the brain. Inhal. Toxicol. 2004;16:437–445. doi: 10.1080/08958370490439597. 

Badman, Ryan P., et al. "Dextran-coated iron oxide nanoparticle-induced nanotoxicity in neuron cultures." Scientific Reports 10.1 (2020): 1-14.

Maher BA, et al. Magnetite pollution nanoparticles in the human brain. Proc. Natl. Acad. Sci. USA. 2016;113:10797–10801. doi: 10.1073/pnas.1605941113.

Yu M, Huang SH, Yu KJ, Clyne AM. Dextran and polymer polyethylene plycol (PEG) coating reduce both 5 and 30 nm iron oxide nanoparticle cytotoxicity in 2D and 3D cell culture. Int. J. Mol. 2012;13:5554–5570.

Huang YW, Cambre M, Lee HJ. The toxicity of nanoparticles depends on multiple molecular and physicochemical mechanisms. Int. J. Mol. 2017 doi: 10.3390/Ijms18122702.

Towards this direction, please include the vendor and item number of the particles you used and flesh out any details you can add about the nanoparticle properties of your nanoparticles versus other comparable work.

Author Response

Dear Reviewer,

We appreciate your comments and suggestions for improvement. Please find the answers to your questions and/or remarks below.

1) The error has been corrected and the figures replaced.

2) If possible, we would like to keep SEM values in the manuscript since construction of a confidence interval assumes knowledge of the probability distribution of the initial data (as a rule, we are talking about a normal distribution). In our case, distribution is not normal, so it seems more appropriate to use SEM to describe the accuracy of estimating the general mean.

3) No, we did not use ANOVA because we compared two groups of animals only. The parameters that we assessed over time were compared only with those observed on week 1. Please advise if we may not use ANOVA in this case.

4) Again, if possible, we would like to keep SEM values. Unfortunately, we cannot say whether MBP fluctuates a lot over time since we did not trace its changes. Thank you for your recommendation on this matter. We shall consider it in our future research.

5) Typo corrected - Table 2 and Figure 5 are now named correctly. The slide showing Fe signals in the control rats has been added. To confirm the absence of interference of biological (essential) iron and/or contamination of the instrument and its components, the EDS spectrum in the control group has been added. The Results section has been supplemented with the following: “Electron dense objects containing Fe were not found in the samples from the control animals; no background signal was detected either.”

The sentence “When determining the degree of damage to the myelin sheath of axons in sections of two regions of the brain, 1,387 unique sites of the myelin sheath of axons were ranked by the presence or absence of damage to its structure.” was replaced by the following: “When determining the degree of damage to the myelin sheath of axons in sections of two regions of the brain, 1,387 unique sites of the myelin sheath of axons (40 to 60 sites × 2 areas of the brain × 7 rats × 2 groups) were ranked by the presence or absence of damage to its structure.”

6) We would like to express our gratitude your comment and valuable articles. We have supplemented the Introduction and Discussions sections of the manuscript with appropriate citations from the latter and highlighted them in yellow.

7) We have added the description of the technique: “Suspensions of Fe2O3 nanoparticles were prepared at the Ural Federal University, Yekaterinburg, Russian Federation, by pulsed laser ablation of thin sheet targets of 99.9 % pure iron in sterile deionized water as described elsewhere (doi.org/10.3390/ijms150712379).”

Round 2

Reviewer 2 Report

Ready for publiaction.